# PROTEIN LANGUAGE MODEL FITNESS IS A MATTER OF PREFERENCE

**Cade Gordon, Amy X. Lu & Pieter Abbeel**
University of California, Berkeley
textttcadegord@berkeley.edu

## ABSTRACT

Leveraging billions of years of evolution, scientists have trained protein language models (pLMs) to understand the sequence and structure space of proteins aiding in the design of more functional proteins. Although they have shown ability to improve efficiency in engineering, it remains unclear if such models capture true biological patterns or artifacts of the training data. We aim to predict the circumstances in which pLMs can successfully perform zero-shot fitness estimation. Our work studies trends observed over hundreds of deep mutational scans across multiple different fitness objectives. We find that the likelihood, or abstractly, implicit preference of a certain protein sequence imbued during pretraining is predictive of fitness prediction capabilities. Both over-preferred and under-preferred wild type sequences harm performance. Using influence functions to causally understand how individual data points increase protein likelihoods, we find that there exists a power law tail due to sequence homology. Lastly, under-performance on low likelihood wild type proteins can be remedied by unsupervised finetuning. These findings that pLM zero-shot fitness estimation can be predicted by the likelihood of the engineered sequence can motivate and improve pLMs' deployment in protein maturation campaigns.

## 1 INTRODUCTION

Protein Language Models (pLMs) have been thought to encapsulate millions of years of evolutionary information through unsupervised pretraining on protein databases. Works have shown that their likelihoods can infer evolutionary trajectories, improve design campaigns, and predict zero-shot mutational effects (Hie et al., 2022; Biswas et al., 2021; Meier et al., 2021). However, more recent works have begun enumerating cases where pLMs likelihoods are influenced by training data compositions that are not direct consequences of natural evolution (Weinstein et al., 2022; Ding & Steinhardt, 2024).

To better understand how training data selection in biological pretraining leads to biases in the learned patterns of pLMs, we propose to study performance through the lens of *preference*. At the individual comparison level, the Bradley-Terry model of preference (Bradley & Terry, 1952) has been shown to be a flexible tool, grounding both modern alignment techniques such as Direct Preference Optimization (Rafailov et al., 2024) and ranking techniques such as ELO. As Ding & Steinhardt (2024) used sequence likelihood to construct species-level ELO scores, we can generalize likelihood to be a measure of sequence preference. By framing our analysis through the lens of preference, we aim to uncover how pLMs implicitly "prefer" certain protein sequences over others. Crucially, we posit that this implicit preference is not just a property of the model, but stems from the preferences encoded in the training data itself. We hypothesize that this multi-layered preference structure, quantified by sequence likelihood, can explain variations in pLM performance across different proteins and tasks.

We start by showing that variations in downstream protein engineering performance can be explained by the likelihood of the starting sequence. To do so, we use deep mutational scan (DMS) datasets to see if underlying log likelihood of starting sequences can be predictive of zero-shot fitness prediction capabilities, and find that **mutation effect prediction on lower likelihood starting (i.e. wild type) sequences have worse performance**, and that **high likelihoods can become harmful after**

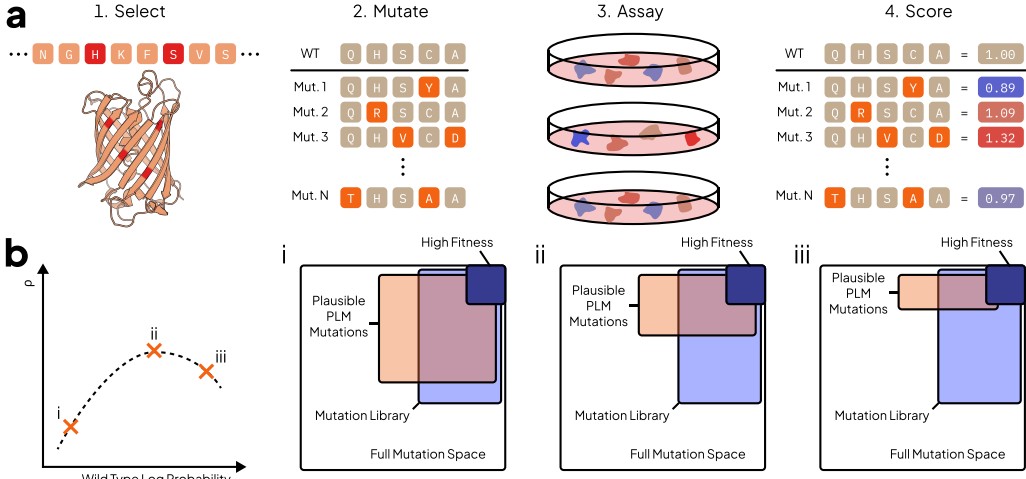

Figure 1: **Understanding DMS and a theory of pLM evolution capabilities.** (**a**) A deep mutational scan follows by choosing a protein then selecting residues to mutate. Mutations are performed, expressed, then assayed for function (e.g. binding, fluorescence, or stability) to determine a fitness score. (**b**) We refine the efficient evolution hypothesis of Hie et al. (2024) suggesting that ability of pLMs for economic protein maturation is dependent on more than nature's plausible mutations. Mutation effect prediction success is reliant on the underlying likelihood or preference towards the non-mutated protein. Ultimately, model performance can be seen as a result of the model's implicit preference and human preference during the data curation pipeline.

**a certain threshold**. From this finding, we generalize the failure pathology from species-level (Ding & Steinhardt, 2024) to individual sequence probabilities. The ability to calculate pseudo log likelihood on datasets of evolutionary-scale magnitude is enabled by **a new derivation of pseudo log likelihood calculation reducing the number of inference passes from $\mathcal{O}(L)$ to $\mathcal{O}(1)$** without any post-training.

A question then emerges: what is causing these sequence likelihoods? We utilize influence functions to understand what proteins from the underlying training dataset increase the likelihood of certain wild type proteins. Our studies reveal that the **distribution of influential data points follow a power law distribution**, and that highly influential sequences can be quickly found using a search tool such as `mmseqs2` (Steinegger & Söding, 2017).

Combining these results, we motivate a past method known as evo-tuning (Alley et al., 2019) to protein designers or test-time training (Sun et al., 2019) to the machine learning community. We leverage our results relating to likelihood to suggest that evo-tuning on low likelihood wild types improves performance and evo-tuning on high likelihood wild types harms performance. This finding matches the intuition garnered from the zero-shot plots and helps to explain counterintuitive dynamics seen in Hsu et al. (2022) that unsupervised finetuning can sometimes worsen performance. Together, our findings improve real-world applicability of increasingly powerful pLMs, also providing theoretical and granular analysis to how homology to training data sequences affects performance.

## 2 RELATED WORKS

**Protein Language Models** Modern pLMs come in two forms: masked or autoregressive models. Of the former, transformer based language modeling efforts like ESM-1B, ESM-1V, and ESM-2 became dominant with BERT-like pretraining objectives (Vaswani et al., 2017; Rives et al., 2021; Meier et al., 2021; Lin et al., 2023). In contrast, ProGen (Alley et al., 2019; Madani et al., 2023; Nijkamp et al., 2023) and other works embraced the GPT-like pretraining style and opted for autoregressive training scheme.

As a result of these training tasks, pLMs have afforded many capabilities surrounding evolution through their likelihoods. Biswas et al. (2021) use pLMs to do property engineering with a limited

number of labeled sequences. Meier et al. (2021) showed that pLMs are capable of zero-shot mutation prediction by comparing the log odds ratios between sequences of interest. Extending this, Hie et al. (2022) proved pLMs can predict the evolutionary trajectory of a range of different proteins and selective pressures. Not only can pLMs predict evolution, Hie et al. (2024) demonstrated that an ensemble of pLMs can improve the affinity of monoclonal antibodies without any information about the antigen.

**Limitations Protein Language Models of Training Data**   Though data research is becoming more and more prevalent within the language modeling, its treatment within the domain of pLMs has not yet been as extensive thus far. Fannjiang & Listgarten (2024) provides an introduction to the relationship between data and model performance in both protein specific and general purpose modeling regimes. On the matter of training as a whole, Weinstein et al. (2022) argued that the density of pLM training data alone doesn't specify the fitness functions of interest, rather due to misspecification pLMs learn to model the stationary distribution enabling their success in fitness tasks. Recently, Ding & Steinhardt (2024) showed how species bias within protein databases has lead to biases in the underlying pretraining sets for most pLMs. It further demonstrated that pLMs fail on design tasks catered towards low-likelihood species and would often revert to over-represented homologs. Lastly, Hermann et al. (2024) uncovered dataset overlaps between commonly used pLM benchmarks and their pretraining datasets. When these overlapping points were removed from the benchmarks, scores decreased.

**Relating Training Data and Model Outputs**   One of the first attempts to relate a probabilistic model's outputs to its input data comes from influence functions (IFs) in robust statistics (Hampel, 1974). Koh & Liang (2017) ported the classic technique to deep learning. A few years later, Grosse et al. (2023) improved computations of IFs enabling evaluation for models of up to 52 billion parameters.

Other modern schools of thought surrounding data have begun understanding downstream model performance as a function of pretraining data for robustness and pruning. Fang et al. (2022) finds that the data distribution is the cause of the large gains in effective robustness for CLIP. Others used data pruning to outperform classical neural scaling laws (Sorscher et al., 2022). With the growing importance being placed on data, tasks are now developing to understand which data points are most important through the DataComp challenges that have been put forward for multimodal and traditional language models (Gadre et al., 2024; Li et al., 2024). Further evidence can be seen with complex data distributions and post-training regimes in works such as Llama 3.1 (Dubey et al., 2024).

## 3   PRELIMINARIES

### 3.1   ZERO-SHOT FITNESS PREDICTION

One of the most common methods for determining how protein fitness changes with mutations is called a deep mutational scan (DMS). As seen in 1a, deep mutational scans start with a protein of interest, then perform combinatorial set of mutations to select residues in the protein. Those are characterized in the wet lab then compared against the unmutated or wild type starter protein.

We choose to focus on masked language models in this study due to their ease of use in point wise mutations and community adoption. Following ESM-1V we utilize, the difference in model likelihoods to compute zero-shot fitness predictions. We denote the wild type sequence as $x$ and the set of mutable residues as $T$. From here we can calculate a predicted gain in fitness $f$ of a mutated sequence $x'$ over $x$ using parameters $\boldsymbol{\theta}$ as the log odds ratio of the mutated sequence against the wild type:

$$f(x', x) = \sum_{t \in T} \log P(y_t = x'_t | x_{\backslash t}, \boldsymbol{\theta}) - \log P(y_t = x_t | x_{\backslash t}, \boldsymbol{\theta}). \tag{1}$$

Each evaluation of $f$ can be compared against real world assay values, thus Spearman correlation can be used as a measurement of agreement (Spearman, 1904) making the task equivalent to ranking. This method is how the predictions in ProteingGym are formulated for the ESM suite of models.

Although this is a powerful method of zero-shot precition, it has two major limitations. First is the linear additivity of fitness. As a result it can't model epistatic fitness interactions. Second is the need

for mutants to be identical in length to the wild type sequence. Since we're evaluating the log odds ratios at certain locations to derive fitness, each residue must have an existing counterpart on both proteins to enable evaluation.

## 3.2 Influence Functions

We find ourselves utilizing a training dataset $\mathcal{D} = \{z_i\}_{i=1}^N$ of $N$ samples and individual sequences $z_i$s. Models are then fit to minimize the empirical risk of a loss function $\mathcal{L}$ to derive an optimal set of parameters:

$$\boldsymbol{\theta}^\star = \arg\min_{\boldsymbol{\theta}^\star} \mathcal{J}(\mathcal{D}, \boldsymbol{\theta}) = \arg\min_{\boldsymbol{\theta}^\star} \frac{1}{N} \sum_{i=1}^N \mathcal{L}(z_i, \boldsymbol{\theta}). \tag{2}$$

In particular, we are interested in understanding the effect of a single point $m$. We choose to weight this data point with some parameter $\epsilon$ arriving at:

$$\boldsymbol{\theta}^\star(\epsilon) = \arg\min_{\boldsymbol{\theta}^\star} \frac{1}{N} \sum_{i=1}^N \mathcal{L}(z_i, \boldsymbol{\theta}) + \epsilon \mathcal{L}(z_m, \boldsymbol{\theta}). \tag{3}$$

Setting $\epsilon = -1$ can be thought of as asking the counterfactual: what if $z_m$ wasn't in the training set? Using a first-order Taylor series of Equation 3, the Implicit Function Theorem, and a few other assumptions we can derive the influence of $z_m$ on $\boldsymbol{\theta}^\star$ in Equation 4. Furthermore, Equation 5 shows that we can calculate the influence of a sequence on a functional evaluation of $f$ via the chain rule.

$$\mathcal{I}_{\boldsymbol{\theta}^\star}(z_m) = \frac{d\boldsymbol{\theta}^\star}{d\epsilon}\Big|_{\epsilon=0} = -\mathbf{H}^{-1}\nabla_{\boldsymbol{\theta}}\mathcal{L}(z_m, \boldsymbol{\theta}^\star) \tag{4}$$

$$\mathcal{I}_f(z_m) = \nabla_{\boldsymbol{\theta}} f(\boldsymbol{\theta}^\star)^\top \mathcal{I}_{\boldsymbol{\theta}^\star}(z_m) = -\nabla_{\boldsymbol{\theta}} f(\boldsymbol{\theta}^\star)^\top \mathbf{H}^{-1}\nabla_{\boldsymbol{\theta}}\mathcal{L}(z_m, \boldsymbol{\theta}^\star) \tag{5}$$

$\mathbf{H}^{-1}$ represents the inverse hessian calculated over $\mathcal{D}$. As one can imagine, calculating this directly becomes prohibitively memory intensive. Grosse et al. (2023) makes it computationally feasible by assuming layer-wise independence and using of EK-FAC for hessian calculation (George et al., 2018). We use the implementation of Bae (2024) in our analysis.

Although giving insight to a very complex relationship, influence functions are not without their limitations. Basu et al. (2020) show the fragility of the method to training alterations like the number of layers, layer width, and the inclusion of weight decay. Extending upon these findings, Bae et al. (2022) showed how a mixture of complications in assumptions and practical training dynamics differ from the idealized construction in influence functions. The work argued that influence functions better capture the proximal Bregman response function, that is what is the effect of removing a data point while also attempting to maintain current predictions?

## 3.3 Pseudo Log Likelihoods

Unlike autoregressive language models, masked language models don't have a natural way to immediately compute the joint likelihood of a sequence. As a result, Wang & Cho (2019) proposed to mask every index of a sequence one-at-a-time then average to derive a PLL (Wang & Cho, 2019):
$\text{PLL}(x) = \frac{1}{L} \sum_{i=1}^L \log P(y_i = x_i | x_{\setminus i}, \boldsymbol{\theta})$.

This formulation suffers from the need to run $\mathcal{O}(L)$ forward passes to compute a perplexity or log likelihood. In response to this, the community only considers autoregressive pLMs when computing fitness values for proteins containing insertions or deletions.

## 4    UTILIZING pLM LIKELIHOODS TO PREDICT ZERO-SHOT SUCCESS

### 4.1    EFFICIENT PSEUDO LOG LIKELIHOOD CALCULATION

For non-autoregressive models, papers (Lin et al., 2023) often use a measure of pseudo log likelihood (PLL) by evaluating $\text{PLL}(x) = \frac{1}{L} \sum_{i=1}^{L} \log P(y_i = x_i | x_{\setminus i}, \boldsymbol{\theta})$. The resultant equation requires $\mathcal{O}(L)$ forward passes to calculate, where $L$ is the length of the protein. Because of the high computational burden, autoregressive language models are sometimes preferred over masked language models when likelihood-based mutation effect prediction is the desired downstream use. To overcome this, Kantroo et al. (2024) perform post training on ESM-2 to predict the distribution of tokens of as if the underlying token of interest had been masked.

We argue that PLL can be approximated for any masked language model in a single forward pass using Algorithm 2 as a result of Remark 4.1. To achieve this, we define the term "mask-consistent", where a token's masked probability is equal to its probability under the implicit assumption that the token might have been scrambled by the training procedure. Using mask consistency, we produce an approximation of the probability of a masked token when the token is not masked during inference in Remark 4.1. As a result, we can calculate PLL for any model without requiring bespoke finetuning or exhaustive resources.

**Remark 4.1** *Under a mask-consistent masked language model with a training scheme that sets training tokens to a random token with probability $\alpha$ and keeps them unchanged with probability $\beta$:*

$$P(y_i = x_i | x_{\setminus i}, \boldsymbol{\theta}) = \frac{\alpha + \beta}{\alpha} P(y_i = x_i | x, \boldsymbol{\theta}) - \frac{\beta}{\alpha}. \quad (6)$$

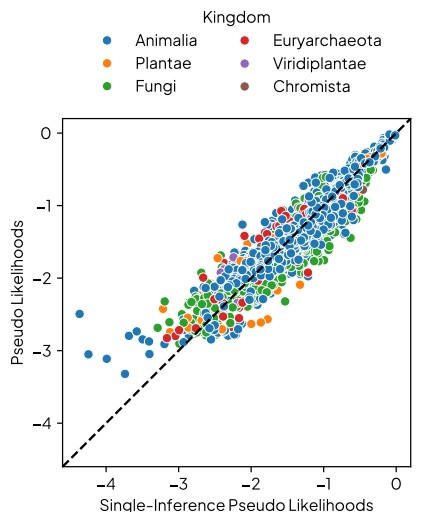

Figure 2: **Single Inference PLL is Consistent with PLL.** ESM-2 650M Single Inference PLL plotted against traditional PLL on a diverse set of proteins.

### 4.2    PROOF OF REMARK 4.1

Recalling the training of BERT (Devlin et al., 2018), 15% of tokens in a sequence are chosen for training. Of those tokens 80% are turned into `[MASK]`, 10% are substituted with random tokens, and the last 10% are left unchanged. We denote the likelihood of substitution with $\alpha$ and the probability of being unchanged with $\beta$.

Letting $\phi | x \sim \text{Bernoulli}(\frac{\alpha}{\alpha + \beta})$ represent the event of a token being a substituted token given that it's not a `[MASK]` token. Using the law of total probability, we can now expand the likelihood of a token being identical to its input token.

$$P(y_i = x_i | x) = P(y_i = x_i, \phi = 0 | x) + P(y_i = x_i, \phi = 1 | x) \quad (7)$$

$$= P(y_i = x_i | \phi = 0, x) P(\phi = 0 | x) + P(y_i = x_i | \phi = 1, x) P(\phi = 1 | x) \quad (8)$$

$$= (1)(\frac{\beta}{\alpha + \beta}) + P(y_i = x_i | \phi = 1, x)(\frac{\alpha}{\alpha + \beta}) \quad (9)$$

$P(y_i = x_i | \phi = 0, x)$ becomes 1 as the token was unchanged. Now the insight comes at the evaluation of $P(y_i = x_i | \phi = 1, x)$. This is the probability of the $i$th token given that the input variable was uninformative. Put in another way, it's the probability if the token was masked. We call the language model "mask-consistent" if $P(y_i = x_i | \phi = 1, x) = P(y_i = x_i | x_{\setminus i})$. Substituting this identity with some algebraic manipulation completes the proof.

$$P(y_i = x_i | x) = \frac{\beta}{\alpha + \beta} + \frac{\alpha}{\alpha + \beta} P(y_i = x_i | x_{\setminus i}) \quad (10)$$

$$\frac{\alpha + \beta}{\alpha} P(y_i = x_i | x) - \frac{\beta}{\alpha} = P(y_i = x_i | x_{\setminus i}) \quad \blacksquare \quad (11)$$

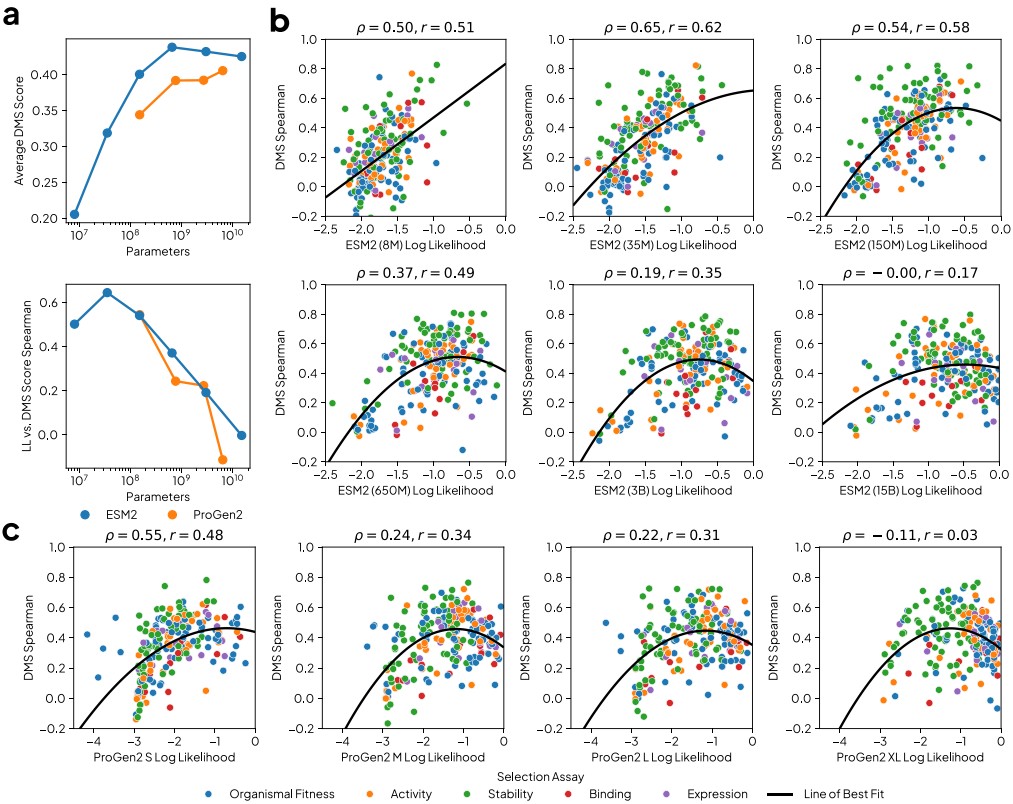

Figure 3: **pLM likelihoods are predictive of mutation effect capabilities.** (**a** Upper) Model parameter counts versus average Spearman correlation on DMS datasets across models. (**a** Lower) Spearman correlation of PLL and performance at each model size. ESM-2 (**b**) and ProGen2 (**c**) of varying scales comparing PLLs DMS wild types against zero-shot performance. Each plots title lists the Spearman ($\rho$) and Pearson ($r$) correlation of the underlying data. Lastly, a second order polynomial regression is fit to the data in all cases resulting in a concave down parabola.

## 4.3 ESM-2 AND PROGEN-2 LIKELIHOODS PREDICT DMS CORRELATION

Our goal is to see if the apparent capacity for pLMs to be used for zero-shot fitness is primarily driven by data or a true understanding of the fitness landscape. To do so, we evaluate pLMs on various deep mutational scan (DMS) tasks, and assess if task performance is correlated with likelihood. Specifically, we take the wild type proteins in 217 DMS studies from ProteinGym (Notin et al., 2023) and calculating PLLs for each of them. In Figure 3 we plot the relationship between PLL and DMS Spearman $\rho$ for the most utilized masked and autoregressive pLMs ESM-2 and ProGen-2.

Generally, ProteinGym task averages improve as parameters increase (Figure 3a upper), with ESM-2 performance degrading past 650 million parameters. But, an inverse scaling law emerges for the Spearman correlation between likelihood and DMS correlation. Figure 3a lower suggests that in small models, a lot of performance can be explained by the likelihood of the wild type sequence, which then degrades as parameters increase across both ESM-2 and ProGen-2.

Further looking at the data in Figures 3b & c illuminates why the correlation between likelihoods and DMS performance decreases. Instead of of likelihoods being less explanatory of zero-shot fitness prediction, higher parameter count models exhibit performance degradation at high likelihoods. This effect is magnified when looking at sequences with probabilities near 1 (or 0 on the log scale) for models such as ESM-2 15 billion and ProGen-2 XL. One interpretation of this phenomenon is that the increased learning capacity of larger models has caused them to overfit on certain regions of sequence space, suggesting that the optimal choice of pretrained model is dependent on the downstream task and dataset.

In all, the results shown in Figure 3 corroborate a theory exemplified in Figure 1b. Low likelihood or under preferred wild type sequences struggle to predict beneficial and harmful mutations. As the likelihood increases so does performance, but after a certain threshold too much preference harms predictive capability.

## 5 UNDERSTANDING pLM LIKELIHOODS USING INFLUENCE FUNCTIONS

Motivated by the predictive phenomenon with likelihoods, we aim to study a more rigorous causal relationship between data and downstream likelihoods using influence functions. We determine a structure of the data involved in these likelihoods and thus preference. In Section 6, we leverage these findings to inspire a method of post-training.

### 5.1 PROTEIN LIKELIHOOD INFLUENCE HAS A POWER LAW TAIL

Influence functions (Hampel, 1974; Koh & Liang, 2017) measure the impact of a training sequence on function of interest. We quantified per-datum influence values in the train dataset to find most influential points on sequence likelihood to investigate our examined phenonmenon. To approximate ESM-2's training distribution, we randomly sample 10,000 proteins from UniRef50 and trim sequences to be of length at most 1,024. We utilize the Kronfluence library (Bae, 2024) to calculate influences conditioning the inverse Hessian on the random samples. Figure 4 depicts

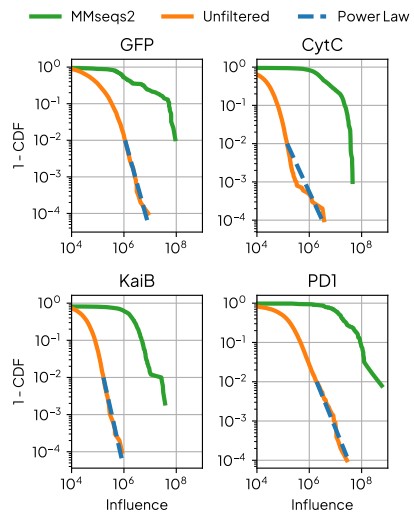

Figure 4: **pLM influence tails exhibits a power law relationship.** ESM-2 650M influences plotted against the complementary cumulative distribution.

influence of these 10,000 points as well as points retrieved by searching UniRef50 using `mmseqs2` on the likelihood four common proteins of interest: Green Fluorescent Protein (GFP), Cytochrome Complex (CytC), KaiB, and Programmed Cell Death Protein 1 (PD1).

Our results suggest two main insights. First, we reproduce the power law tail observed for traditional LMs in Grosse et al. (2023), suggesting similar data dynamics between pLMs and LMs. From a biological perspective, Qin & Colwell (2018) finds a power law tail in the covariance of phylogenetic protein systems, which might lend a way to understand this result. Second, for each of the four proteins, `mmseqs2` found some of the most influential proteins when compared to an unfiltered set. This means that search might serve as an efficient way to deduce which training samples can be used to improve performance.

### 5.2 INFLUENCE DIMINISHES WITH EDIT DISTANCE

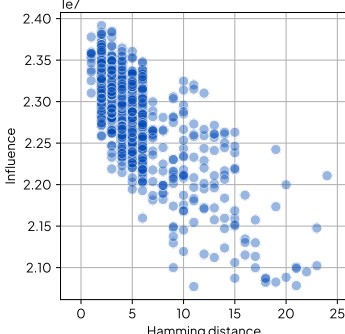

Figure 5: **Influence on likelihood decreases as hamming distance increases.** The diminishing influence sugggests that homologous sequences have a the greatest causal link to preference.

As protein search yields many of the influential proteins, a natural thought would be that influence is related to the amount of homology between a training data point and the sequence whose likelihood is in question. To investigate this phenomenon, we selected DMS studies by the maximal number of edits from the wild type sequence taking the top 10 from ProteingGym. From each DMS study, a random 1,000 samples are used as a synthetic set of training examples to examine what would happen were a protein with some number of mutants from the wild type within the train set.

We present a representative example in Figure 5 using ESM-2 650M on Sinai et al. (2021)'s DMS

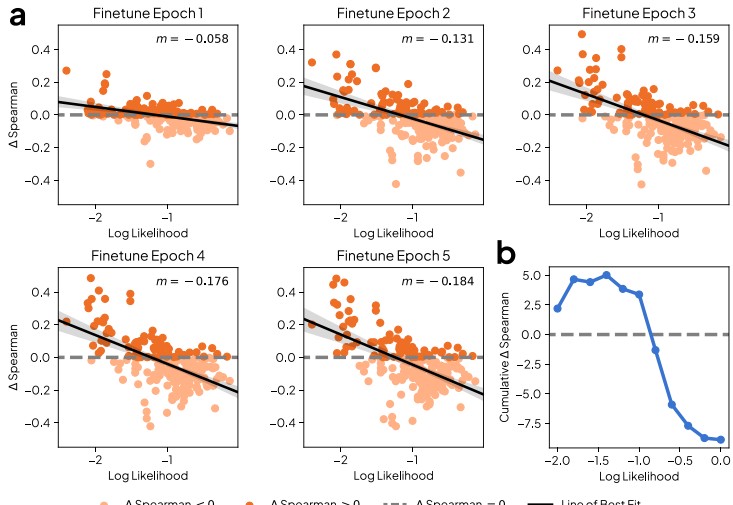

Figure 6: **Wild type likelihoods are predictive of finetuning success.** (**a**) The change in Spearman correlation on 217 DMS studies over 5 epochs. Each plots upper right corner denotes the slope of a linear model. (**b**) The cumulative gain in performance for studies below likelihood $\epsilon$.

| Model Name | Selection Criteria | | | | | | |
| | Activity | Binding | Expression | Organismal Fitness | Stability | Mean | Weighted Mean |
|---|---|---|---|---|---|---|---|
| ESM-2 650M | 0.441 | 0.327 | 0.415 | 0.390 | 0.523 | 0.439 | 0.419 |
| + Finetune $\epsilon = -1.4$ | $0.461^{0.020}$ | $0.356^{0.029}$ | $0.437^{0.022}$ | $0.425^{0.035}$ | $0.534^{0.011}$ | $0.462^{0.023}$ | $0.443^{0.024}$ |
| + Finetune $\epsilon = 0$ | $0.391^{0.050}$ | $0.271^{0.056}$ | $0.393^{0.022}$ | $0.385^{0.005}$ | $0.444^{0.079}$ | $0.398^{0.041}$ | $0.377^{0.042}$ |
| ProGen2 XL | 0.404 | 0.291 | 0.418 | 0.389 | 0.445 | 0.406 | 0.389 |
| EVE | 0.464 | 0.354 | 0.404 | 0.449 | 0.487 | 0.454 | 0.432 |
| MSA Transformer | 0.459 | 0.32 | 0.435 | 0.416 | 0.476 | 0.438 | 0.421 |
| TranceptEVE L | 0.492 | 0.359 | 0.457 | 0.466 | 0.5 | 0.474 | 0.455 |

Table 1: **Comparing finetuned ESM-2 650M to state-of-the-art pLMs.** ESM-2 650M with and without finetuning at various values of $\epsilon$ compared to other fitness prediction models. Results for all non ESM-2 models are reported from ProteinGym. In line with earlier findings that high likelihoods can harm performance, we find that applying a threshold to which sequences we finetune on improves performance.

data chosen on the basis of having the second highest edit distance and a fairly uniform edit distance distribution. The influence of a mutant protein against its wild type decreases as a function of its edit distance to the wild type. Further evidence can be seen in Appendix Figures 8 and 9. Across five different ESM-2 model scales (8M, 35M, 150M, 650M, and 3B) and all 10 datasets, we find that as edit distance increases influence decreases. Variations in the clarity of this relationship might be explained by the number of locations mutated on the protein of interest and underlying distribution of edit distances.

## 6 EVO-TUNING PLMS TO IMPROVE FITNESS PREDICTION

Motivated by the finding that that low probability sequences underperform, and that `mmseqs2` serves as a simple heuristic for finding likelihood-influencing data points, we propose a simple remedy: a pLM can be finetuned to increase the likelihood of the wild type sample for a protein engineering task, but sequences with sufficiently high likelihoods should remain unchanged.

We perform unsupervised finetuning on homologous sequences (sometimes referred to as evo-tuning (Alley et al., 2019; Biswas et al., 2021)) of ESM-2 650M on a set of sequences derived by searching

UniRef100 for the wild type sequence on each of the 217 DMS studies in ProteinGym separately. Post training utilizes AdamW (Loshchilov et al., 2017) with a learning rate of `1e-6` for 5 epochs on the 1,000 most similar proteins to wild type as determined by E-value of `mmseqs2` search with a maximum cut off of 1. Finetuning starts at a batch size of 32 is progressively halved in the occurence of an out of memory exception. Each run consumes a single 80GB A100 GPU. As our findings above further indicate that too much likelihood harms performance, we only consider finetuning models where the log likelihood falls below a threshold $\epsilon$.

Examining the change in Spearman correlation on the DMS test bed from finetuning with respect to initial sequence log likelihood in Figure 6, performance improvement is anti-correlated with starting probability. Low likelihood sequences benefit from training, while high likelihood sequences get harmed, an effect that gets amplified as more training occurs. This effect further corroborates the model shown in Figure 1b. The nontrivial performance harm of too much likelihood might be explained through Weinstein et al. (2022)'s findings that modeling the sample data distribution density isn't what leads to fitness prediction. In our case, further finetuning on high likelihood regions in sequence space might cause memorization of phylogenetic artifacts instead of fitness signals.

If the model is naively finetuned before evaluation on every DMS, then performance degrades as seen in Table 1 at $\epsilon = 0$, in accordance with Meier et al. (2021). Figure 6b investigates how only performing finetuning on samples that fall below some likelihood threshold $\epsilon$ leads to a gain in total correlation accross ProteinGym. $\epsilon$ is evaluated at 11 equally spaced log likelihoods from $-2$ to $0$. The best performance is observed at $\epsilon = -1.4$. After accounting for this procedure, ESM-2's scores jump and outperform models that utilize evolutionary information through MSAs like EVE (single) and MSA Transformer (single) (Frazer et al., 2021; Rao et al., 2021), while becoming competetive with the state-of-the-art hybrid MSA and pLM model TranceptionEVE (Notin et al., 2022).

## 7 DISCUSSION

In this work, we proposed studying pLMs through the framework of preference. Our findings demonstrate that sequence likelihood, which can be thought of as implicit preference, can predict pLM performance on a diverse set of fitness prediction tasks. In thinking about pLM capabilities through preference, we illuminate different layers of preference suggesting that the learned pLM behavior is reflective of the user level bias in curation of training data, and may not always reflect evolutionary patterns as one might assume.

We show that both low and high likelihood sequences suffer in performance, suggesting that over or under preferring data is harmful for fitness prediction capabilities. By utilizing influence functions, we're able to link the observed likelihoods to training data in a causal manner suggesting that homologous protein training data is most responsible for the driving the latent preferences of these models. Combining these two findings we arrive at a nuanced way to improve off the shelf models: unsupervised finetuning on regions of low-likelihood space.

Crucially, our work demonstrates that the ability of a pLM to predict fitness is indeed a matter of preference - not just of the model, but of the human choices and resource limitations that shape data curation. This perspective helps us understand when these models will succeed or fail in applied use giving us ways to improve their success.

Looking forward, our findings motivate the need to think more intentionally about pretraining and evaluation. Naive usage of sequence databases and scaling will magnify the biases training data leading to miscalibrated preferences. Researchers should curate the next generation of datasets and supervision schemes to adjust for this. Lastly, understanding the interplay between data and performance suggests that researchers should ensure proper test task representation during pretraining when benchmarking as suggested by Dominguez-Olmedo et al. (2024). Algorithmic differences might be overshadowed by human preferences at the data level confounding whether a model better captures the biology of proteome.

In conclusion, by thinking about pLM behavior in terms of preference, we've provided a new lens through which to understand and improve these powerful tools. As the field of protein engineering continues to evolve, studying pLMs through their likelihoods and training distriubtions offers a promising path to diagnose and train more effective protein language models.

ACKNOWLEDGEMENTS

We're grateful for our conversations with Cem Anil surrounding influence functions, and Frances Ding surrounding bias in pLMs. We also thank Seyone Chithrananda and Ludwig Schmidt for their feedback on this work. Our experiments are enabled by the resources donated by NVIDIA. Cade Gordon is sponsored by the Siebel Foundation. Amy X. Lu acknowledges support from NSERC PGS D. Pieter Abbeel holds concurrent appointments as a Professor at UC Berkeley and as an Amazon Scholar. This paper describes work performed at UC Berkeley and is not associated with Amazon.

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

## APPENDIX A   FURTHER EXPERIMENTAL DETAILS

### A.1   PROTEIN SEARCH

We utilize the July 24, 2024 UniProt releases of both UniRef50 and UniRef100 (Suzek et al., 2015). The search performed over UniRef50 used the default sensitivity and a maximum E-value of 1 to be inline with Lin et al. (2023). Similarly, searches on UniRef100 utilized a maximum E-value of 1, but also a sensitivity of 7.5.

### A.2   INFLUENCE FUNCTIONS

Using the Kronfluence (Bae, 2024) we calculate influence functions using a method identical to that introduced in Grosse et al. (2023). As such the computation has three major components that can be altered: the training loss function $\mathcal{L}$, the measurable value $f$, and the data used to condition the Inverse Hessian $\mathbf{H}^{-1}$.

For both the loss function and measurable function we want something that mirrors masked langauge model training and PLL calculation. To that end, we choose the function $\frac{1}{L}\sum_{i=1}^{L}\log P(y_i = x_i|x,\theta)$. This function is both a rough estimate of the training objective, while removing the need for a stochastic loss estimation of a certain point and is similar to PLL. Although, we do introduce another version of PLL that requires only a single inference, the clipping would result in certain tokens not contributing to the loss. As a result, this form both avoids stochasticity and multiple inference passes to arrive at both our training loss and observed function PLL.

As the Inverse Hessian will approximate some of the curvature of our data, we want to choose conditions that match our study of interest. As a result, we used two different setups to perform this training data conditioning. In Section 5.1, we use 10,000 randomly sampled points from UniRef50 as can be thought of as an estimate of ESM-2's training data. Then in Section 5.2, we condition on each individual study's downstream points as those will help capture the local curvature of the likelihoods around the wild type protein of interest.

## APPENDIX B   SINGLE-INFERENCE PSEUDO LOG LIKELIHOOD

In this section, we expand on the implications of Remark 4.1. First we use it to propose Algorithm 2 that leverages this result to reduce the number of inferences from $\mathcal{O}(L)$ to $\mathcal{O}(1)$. Lastly, to ensure that the assumptions weren't too lenient, we evaluate our likelihoods on $7,545$ sequences of varying species and length showing tight agreement between our Single-Inference PLL and traditional PLL.

Though Kantroo et al. (2024) also provide a method for enabling PLL calculation in a single pass, it requires training a separate neural network ensemble to estimate the masked quantities. Hence, whenever one wants to calculate PLL in one pass for any new model this adapter ensemble must be trained before PLL calculation. Our method bypasses this need, letting PLL be evaluated out of the box in a single inference pass.

### B.1   ALGORITHM

Algorithm 1 provides detail on the traditional method of PLL calculation. Each residue is masked one at a time, then the log probabilities of being the inputted sequence at the masked location are averaged into a single scalar.

---

**Algorithm 1** Traditional Pseudo Log Likelihood Calculation

---

**Require:** $x \in \mathcal{V}^L$
1:  $z \leftarrow 0$                                                                 ▷ Initialize PLL
2:  **for** $i \in \{1, \ldots, L\}$ **do**
3:      $z \leftarrow z + \frac{1}{L}\log P(y_i = x_i|x_{\setminus i})$               ▷ Mask and infer
4:  **end for**
5:  **return** $z$

---

To overcome the for loop, and thus $\mathcal{O}(L)$ forward passes, Algorithm 2 relies on Remark 4.1 to derive probability values equivalent to the masked probabilities. As each probability of interest is captured in a single forward pass, we can side step the for loop now only requiring $\mathcal{O}(1)$ or a single forward pass to calculate PLL. One limitation becomes apparent in calculating our closed form probability of interest, it's plausible that $\frac{\alpha+\beta}{\alpha}P(y_i = x_i|x) < \frac{\beta}{\alpha}$ leading to a negative probability. Since probability must be a non-negative measure, we therefore clip values that are too low to some $\epsilon$.

---

**Algorithm 2** Single-Inference Pseudo Log Likelihood

---

**Require:** $x \in \mathcal{V}^L, \epsilon \in \mathbb{R}^+$
1: $p \leftarrow P(y = x|x)$           $\triangleright$ Perform inference once, $p \in [0,1]^{L \times \mathcal{V}}$
2: $p' \leftarrow \max(\frac{\alpha+\beta}{\alpha}p - \frac{\beta}{\alpha}, \epsilon)$        $\triangleright$ Use Thm. 4.1 and $\epsilon$ for negative probabilities
3: **return** $\frac{1}{L}\sum_{i=1}^{L} \log p'(y_i = x_i)$       $\triangleright$ Vector sum and in place operation

---

To ground $\alpha$ and $\beta$ for the reader, BERT plus the ESM family use 0.1 for both $\alpha$ and $\beta$. For each token included in loss calculation, the models substitute and hold each token with 10% chance given. We utilize this value for all calculations of Single-Inference PLL within this study.

### B.2 EMPIRICAL VALIDATION OF SINGLE-INFERENCE PSEUDO LOG LIKELIHOOD

To assess the correctness of the Single-Inference PLL calculation, we seek to measure its correspondence with classic PLL calculation. We calculate PLL using both methods using $7,545$ of various lengths, species, and families from Ding & Steinhardt (2024). In Figure B.2 we can see the calibration of the two quantities against one another.

The two methods have strong correlation statistics. Calculating the correlation between PLL and Single-Inference PLL yields Spearman $\rho = 0.923$ and Pearson $r = 0.930$. The $P$-values for both underflow the range of Python's floating point precision. Spearman correlation was seen to slightly improve as $\epsilon$ increased while Pearson correlation would lessen. As our interests lie in rank order statistics, we chose to utilize an $\epsilon = 10^{-3}$ to have more rank agreement during the studies of the main text.

## APPENDIX C    EXTENDED INFLUENCE FUNCTION PLOTS

We plot the relationship between influence and edit distance for the ESM-2 family at scales between 8 million and 3 billion. Each plot depicts a single model examining the effect of adding the mutants of a DMS study into the underlying training set. We selected the 10 studies with the highest maximal edit distance from their wild type sequence.

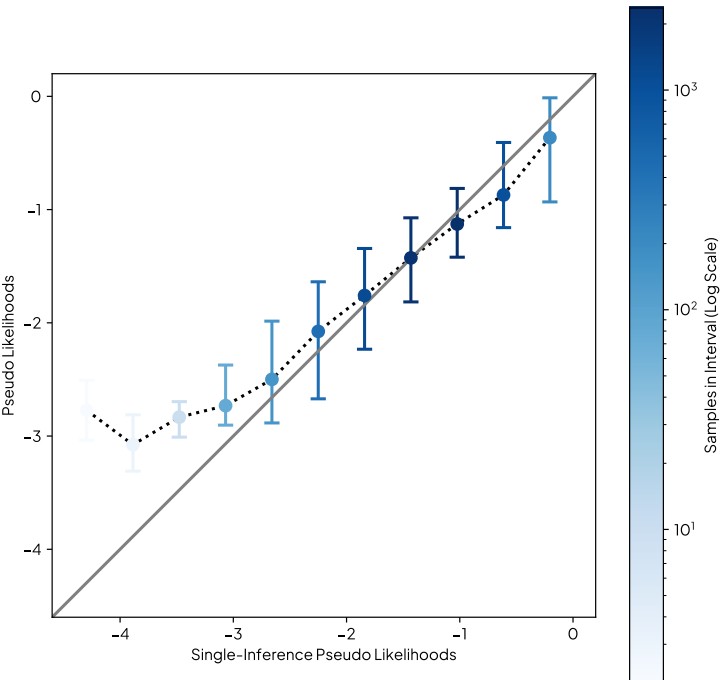

Figure 7: **Single-Inference PLL is calibrated with PLL on a diverse set of proteins.** We plot the mean and inner 95% quantile for 11 evenly spaced bins between -4.5 and 0. Each data point is colored by the number of samples used to derive the statistics for the bin. Although low likelihood sequences seem inconsistent, there are few of these points in the dataset making it hard to draw a confident conclusion on their calibration. On the other hand, for the bins with greater sequence counts, Single Inference PLL aligns well with with traditional PLL.

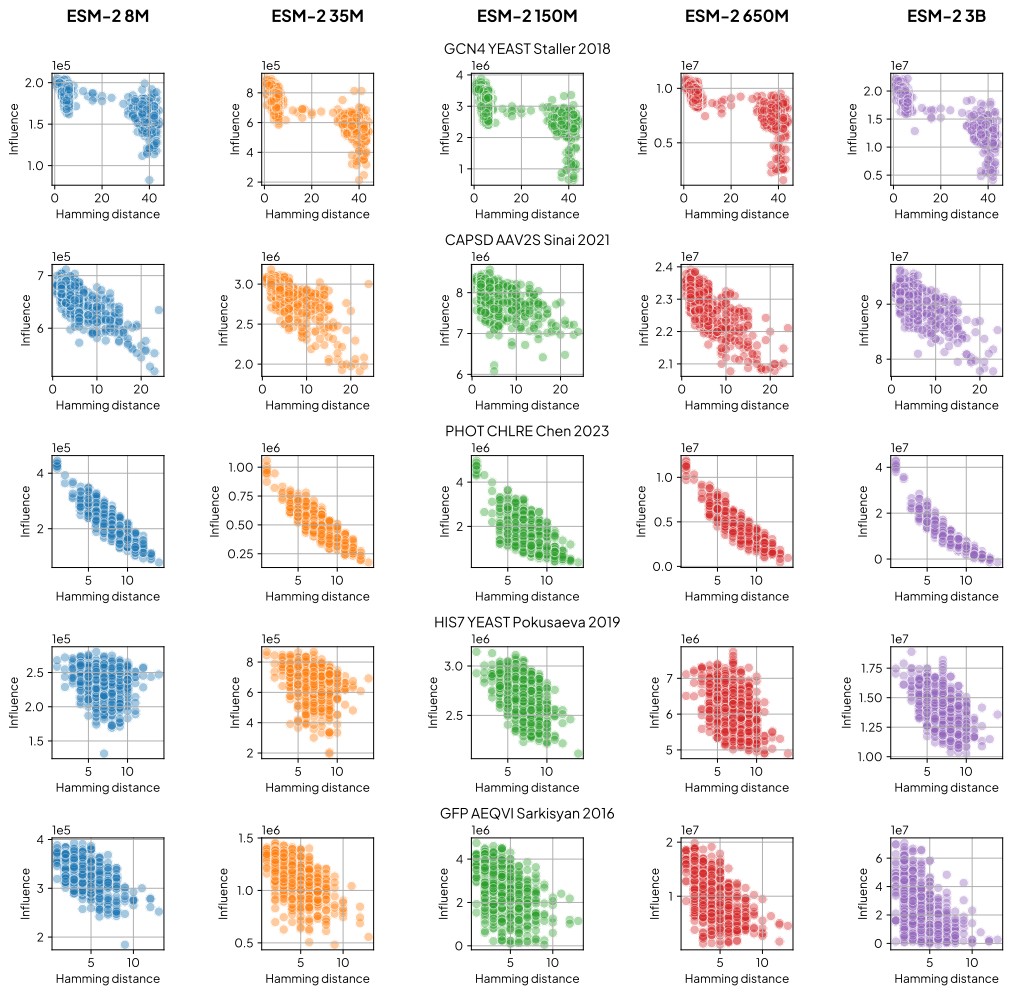

Figure 8: **Influence versus edit distance for the top 5 DMS studies with the most mutations from wild type.** Each row represents a unique study an each column an ESM-2 Model 8M, 35M, 150M, 650M, and 3B (left to right).

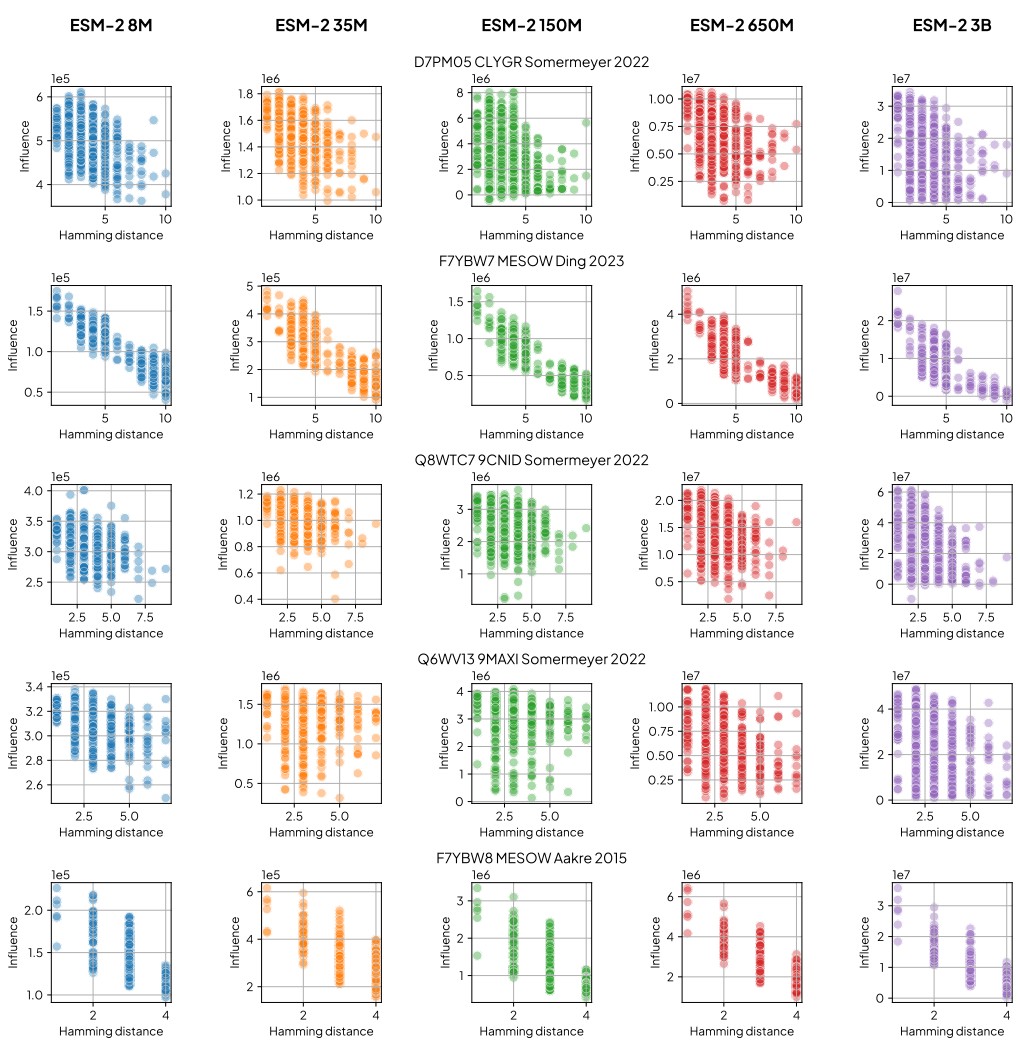

Figure 9: **Influence versus edit distance for the top 6-10 DMS studies with the most mutations from wild type.** Each row represents a unique study an each column an ESM-2 Model 8M, 35M, 150M, 650M, and 3B (left to right).

