# OpenReview forum: "Protein Language Model Fitness is a Matter of Preference"
_ICLR.cc/2025/Conference — ICLR 2025 Poster_

### Official Review · Reviewer_HjzH · 2024-10-18

**Soundness:** 3
**Presentation:** 3
**Contribution:** 3
**Rating:** 6
**Confidence:** 4

**Summary:**

This paper examines how protein language models (pLMs) perform in predicting protein fitness, specifically zero-shot fitness estimation. It argues that a model's success in such tasks can be predicted by the likelihood or implicit preference of a protein sequence learned during pretraining. The study finds that both high and low preferences for a sequence can negatively impact prediction accuracy, but using influence functions to trace these preferences back to specific training data reveals a power law distribution in influential data points. The research suggests fine-tuning on underrepresented sequences as an approach to improving prediction capabilities and provides insights into how training data shapes model performance, guiding more effective use of pLMs for protein engineering.

**Strengths:**

1) The paper offers a perspective by framing pLM behavior in terms of implicit preferences derived from training data, shedding light on limitations and potential improvements for zero-shot fitness prediction.

2) The paper shows a rigorous analysis of pLM performance across multiple datasets and uses influence functions effectively to understand the causal effects of training data.

3) The structure is clear, with theoretical explanations, empirical validations, and practical recommendations for pLM deployment in protein engineering.

**Weaknesses:**

1) The reliance on influence functions has known limitations, such as sensitivity to model hyperparameters, which could affect the validity of conclusions.

2) The paper heavily focuses on likelihood as a key predictor of performance without sufficiently exploring alternative metrics or features that may also contribute to zero-shot success.

3) The paper's experimental setup could be enhanced by including more diverse datasets or varying conditions to test the robustness of the evo-tuning technique.

**Questions:**

1. Could you elaborate on how the choice of loss function and measurable value impacts the influence function results? Specific examples could enhance understanding.

2. Could the authors provide a more extensive comparison across different architectures (e.g., autoregressive vs. masked models) to see if the observed trends are generalizable?

3.  Would results vary significantly with different pretraining datasets, especially those curated with diverse or balanced sequence representation?

4. Have the authors considered methods to explicitly model epistatic interactions in conjunction with the proposed likelihood-based framework? This could potentially address the limitations associated with linear fitness predictions.

---

> ### Author Response · Authors · 2024-11-21
> **Response to Reviewer HjzH**
>
> We appreciate your time taken to engage with the work. Thank you for noting the new perspective we frame fitness with, the paper’s rigor in the derivation of claims, and clear structure. Below we aim to address the questions raised:
>
> **“Could you elaborate on how the choice of loss function and measurable value impacts the influence function results? Specific examples could enhance understanding.”**
>
> We will update our work by adding the following statement, “When calculating influence functions, the loss function must match that used at training time, as this ensures the influence calculations reflect the actual optimization landscape the model encountered during training. For example, in language models this is the cross-entropy loss and for regression models this could be mean squared error. The measured value computed from the model's prediction may be any arbitrary differentiable function, allowing us to analyze different aspects of model behavior. In our analysis, we chose likelihood as our measured value because of its predictive capability towards zero-shot fitness calculation. This choice helps us identify which training examples most strongly lead to our findings in Section 4.3.”
>
> **“Could the authors provide a more extensive comparison across different architectures (e.g., autoregressive vs. masked models) to see if the observed trends are generalizable?”**
>
> This is an important question and we aimed to address the differences in architecture by showing multiple model scales for commonly used masked and autoregressive models in Figure 2b and 2c respectively.
>
> **“Would results vary significantly with different pretraining datasets, especially those curated with diverse or balanced sequence representation?”**
>
> We also find this question very interesting and the next natural question to extend on this work. As we do not have the resources to train a model on well curated balanced data, we leave this as a question for future research. We conjecture that a balanced dataset would still exhibit power laws over data influence, but would hopefully see less low and high likelihood sequences improving overall performance.
>
> **“Have the authors considered methods to explicitly model epistatic interactions in conjunction with the proposed likelihood-based framework? This could potentially address the limitations associated with linear fitness predictions.”**
>
> This is an important future direction for the work. Most of the DMS substitution datasets that we utilized often rely on only single mutants, making the linear fitness prediction a successful metric on this test bed. Kantroo et al. 2024 [1], which proposes a single forward pass PLL metric relying on a finetuned ensemble, gets towards this ambition by using the entire sequences PLL to calculate a fitness score, but ultimately found it to underperform linear fitness prediction.
>
> [1] Kantroo et al. 2024, “Pseudo-perplexity in One Fell Swoop for Protein Fitness Estimation”

---

> > ### Comment · Reviewer_HjzH · 2024-11-26
> >
> > Thank you for the reply. I keep my score.

---

### Official Review · Reviewer_LN23 · 2024-10-27

**Soundness:** 3
**Presentation:** 4
**Contribution:** 4
**Rating:** 8
**Confidence:** 4

**Summary:**

The paper investigates how protein language models (pLMs), which learn from evolutionary data, can predict protein fitness in zero-shot scenarios. The study highlights that the likelihood or preference for certain protein sequences, as captured during pretraining, significantly affects the performance of these models. Key findings are as follows. (1) High or low sequence likelihoods can both impair the predictive performance of pLMs. Sequences with moderate likelihoods perform better in predicting mutation effects. (2) The training data's composition affects sequence likelihoods, and influence functions reveal that the impact of individual data points follows a power law, with homology playing a key role in driving model preference. (3) Fine-tuning (evo-tuning) on homologous sequences can enhance performance for low-likelihood sequences but may degrade it for high-likelihood sequences. This approach addresses the limitations in traditional pLM deployment.

**Strengths:**

1-The paper provides a detailed investigation into the performance of pLMs across various fitness prediction tasks, utilizing data from hundreds of deep mutational scans. This breadth of analysis allows for a more generalizable understanding of pLM behavior, as it encompasses diverse proteins and fitness objectives. By examining both high- and low-likelihood sequences, the study offers insights into the conditions under which pLMs perform well or poorly, making the findings applicable to a wide range of protein engineering challenges.

2-The study employs influence functions to causally link training data with model outputs, uncovering how individual training sequences impact the likelihood assigned to protein sequences. By showing that the influence follows a power law distribution and is closely tied to sequence homology, the paper offers a way to identify critical data points that drive model preferences, guiding the selection of training data and finetuning strategies.

**Weaknesses:**

The study investigates only two models, ESM-2 and ProGen-2, which, while both powerful and relevant, represent a narrow slice of the available landscape of protein and sequence-based models. Including a broader set of models, such as ProtBERT, ProtGPT-2, could lend further robustness to the findings. Expanding the selection of models would provide a more comprehensive assessment and potentially lead to stronger, more generalized conclusions about model performance across biological tasks.

Line 230, Section 4.2 is not at the Appendix.

**Questions:**

Line 347, why can only 10K protein sequences from UniRef50 approximate the distribution of training data of EMS-2?

---

> ### Author Response · Authors · 2024-11-21
> **Response to Reviewer LN23**
>
> Thank you for your review and feedback on the work. We appreciate you noting the generalizability of our analysis and the role of influence functions to causally link behaviors to training data. Below we detail our response to your questions and feedback.
>
> We’ll make sure to fix line 230 to reflect Section 4.2 being in the main body of the text. Thank you for catching this.
>
> **“Line 347, why can only 10K protein sequences from UniRef50 approximate the distribution of training data of EMS-2?”**
>
> While 10K samples is much smaller than the entire sequence distribution of UniRef50, our goal was to understand the broad scale distribution of influences across the training dataset so this value was chosen as it both has sufficient sequences to explore many different parts of sequence space while being resource aware. Calculating influence functions in itself is quite expensive, requiring the estimation of an inverse hessian product that takes a significant amount of memory and FLOPs. We believe this value to be large enough to capture our distribution of interest fairly in a tractable manner and examine influence distributions against other subsets of identical size for fairness.

---

> ### Comment · Reviewer_LN23 · 2024-11-29
> **Commys by Reviewer LN23**
>
> I read the response and thank the authors for their rebuttal.

---

### Official Review · Reviewer_NtpF · 2024-10-31

**Soundness:** 3
**Presentation:** 2
**Contribution:** 3
**Rating:** 6
**Confidence:** 2

**Summary:**

The paper studies masked language models used for zero-shot prediction of the effect on fitness of a mutation to a protein. It argues that biases in protein LM pretraining data may result in a skew in the model's likelihood scores, which are observed to correlate with a model's zero-shot fitness prediction accuracy. The paper draws on influence functions to motivate revisiting an old fine tuning technique (evo-tuning) and shows that this increases performance on a standard suite of deep mutational scanning (DMS) datasets.

**Strengths:**

The paper's core observation, that there is a correlation between the overall likelihood of a protein sequence and the accuracy of zero-shot fitness prediction for variants of the sequence, is interesting. I think a number of proteins+ML researchers will be inspired by this to think about potential solutions.

The paper offers a concrete modeling suggestion that provides mild improvements on the DMS datasets. The technique, evo-tuning, was contributed in prior work, however. It just hadn't been revisited for these DMS problems.

**Weaknesses:**

I raise a number of questions below. My chief concerns are that (1) the presentation on a new algorithm for constant-time computation of sequence likelihood has some serious flaws, the (2) influence function analysis occupies a lot of space in the paper and relates to some cool recent work, but doesn't drive the paper's narrative enough to being included, and (3) the overall story about 'preferences' could be reframed in terms of a basic story about underfitting vs. overfitting during pretraining.

**Questions:**

======
The U-shaped relationship between fitness prediction accuracy and sequence likelihood is interesting, but I struggled to see a concrete argument about why this is reflecting some specific issue regarding the structure/balance of protein LM pretraining datasets instead of just a basic impact of undertraining/overtraining. For wildtype sequences with low likelihood, these may come from regions in sequence space with low frequency in the pretrianing data, which means that model is under-trained for this part of sequence space, and similarly it is over-trained for sequences with high likelihood.  Perhaps I'm missing something. What part of this observation is specific to proteins and why does it rely on a notion of 'preference'?

I understand that 'preference' is a trendy word due to things like DPO and ELO, but in this paper I found it to be a fairly lightweight rephrasing of the likelihood of a sequence. There is no notion of external supervised preference data (like DPO uses) or in some global ordering like in ELO. In Ding et al, 'preference' was important, because it provided a way to compare neighboring protein sequences, and the paper had a clear story about a confounding factor (species) that drives this preference.

======
I found sections 4.1 and 4.2 (on predicting the pseudo-likelihood of a sequence in a single pass) to be highly confusing.

First of all, I found the theorem statement to be poorly defined. It says "Under a mask-consistent masked language model that sets masked tokens to a random token with probability alpha and keeps them unchanged with probability beta…". A model doesn't 'set' a token. Also, what does it mean for a model to 'keep' a token 'unchanged.' Do you mean something along the lines of "given a masked language model that was trained on data with the following distribution…"?
Finally, you need to define what 'mask consistent' means in the actual theorem statement.

Besides the details of how the theorem is presented, I actually don't understand the core statement either. Overall, the statement is of the form for 'task of computing quantity A, we have a new algorithm that is O(1), whereas prior methods for computing A were O(L)'. However, what exactly is A and do the new and old approaches provide the exact same output for all possible models, or only when models have maximized the likelihood of training data from a particular distribution?

My impression is that the baseline O(L) approach would compute the pseudolikelihood by taking L forward passes of the model, where each pass replaces the ith token in X with [MASK] in order to compute the probability of x[i] given the rest of X. However, if I understand correctly, your approach passes X into the model with no mask. Given that the actual inputs to the model are different, which means that the activations are different, it appears to me that the approaches would provide numerically different outputs, albeit they may be similar if models have been trained for sufficiently long on a certain training distribution. I'm assuming that this is what is reflected in the lack of perfect correlation in Fig 2. This level nuance is certainly not present elsewhere in the text of sec 4.1 and 4.2.

Also, why is the single-pass approach actually necessary in your experiments? In my understanding, there are roughly a million distinct protein sequences in ProteinGym, so you would need to do one million forward passes of the model. Given that you had the compute resources to independently evo-tune the model on 200+ datasets, I'm assuming that one million forward passes was doable.

Also, you should add a discussion of the relationship between your technique and  "Predicting a Protein’s Stability under a Million Mutations."

 ======
I found the discussion of protein 'fitness' to be too vague. The paper focuses on DMS datasets that measure different attributes of a protein, such as stability, binding, etc, yet it uses assay-independent models based on evolution that predict a single number for a protein sequence, regardless of what attribute of the protein is being measured. Therefore, there is a fundamental ceiling on DMS accuracy for these assay-independent models, since different attributes of a protein can be anti-correlated in practice (e.g. an enzyme's catalytic activity and stability).

Note that different types of fitness functions may be easier to predict using ML than others (e.g., I can see an argument that stability would be easier to predict than catalytic activity). This speaks to a potential confounder in your analysis: there is a correlation between the protein sequence's content and what kinds of DMS studies it appears in. What if, for example, the correlations you are seeing are due to the fact that low-likelihood proteins are from the sort of species where the DMS datasets are based on high-level observations about organismal survival, instead of low-level molecular biology measurements like binding? Perhaps predicting organismal survival is a fundamentally more difficult task.

======
I think influence functions are cool and I enjoyed seeing that you used them to study protein LMs, but I don't think the inclusion of the influence function results drives the paper's story enough to be included. Section 5.1 just demonstrates that the established power-law distribution of influence function values appears for protein LMs (extending prior work doing this for other LMs. I found these sentences confusing: First, we reproduce the power law tail observed for traditional LMs in Grosse et al. (2023), suggesting similar data dynamics between pLMs and LMs. From a biological perspective, Qin & Colwell (2018) finds a power law tail in the covariance of phylogenetic protein systems, which might lend a way to understand this result." Can you please explain the connection to Qin and Colwell in more detail? Besides the fact that the paper is about proteins and discusses power laws, is there more of a connection?

Can you elaborate on the result of sec 5.2? How is this observation actionable? How does it inform the other parts of the paper?

====
I didn't understand figure 1B. Points i, ii, and iii on the left reflect proteins with different likelihood scores for the wildtype. On the right, these have different sized boxes for the set of 'plausible' mutations. I'm assuming the set of plausible mutations is the set of mutations that don't significantly decrease the sequence's likelihood. Why would the overall likelihood of the wildtype be correlated with the size of this set?


====Update after the author's response period====
I appreciate the healthy back-and-forth we had. The new framing of your approximation algorithm is much more sound. I have raised my score. If the paper is accepted, there is not official mechanism to enforce that you change the exposition to reflect our discussion. However, it would be greatly appreciated that you switch to the new exposition, as it is more technically correct (in the sense that it is up-front about the approximation being an approximation), and more user-friendly to readers.

---

> ### Comment · Reviewer_NtpF · 2024-11-19
> **No response to my comments so far**
>
> I raised a number of concrete questions in my review. Can you please respond to them?
>
> Note that many of the other reviewers gave positives scores for the paper, but their reviews were fairly short and high-level. It should be treated as gating for paper acceptance to respond to my review.

---

> ### Author Response · Authors · 2024-11-21
> **Response to Reviewer NtpF Part 1**
>
> We appreciate the thorough review of this paper, and have attempted to address each of the points raised in the comment below. The feedback has assisted in creating better writing around key elements of the work.
>
> **“I struggled to see a concrete argument about why this is reflecting some specific issue regarding the structure/balance of protein LM pretraining datasets instead of just a basic impact of undertraining/overtraining. [...] What part of this observation is specific to proteins and why does it rely on a notion of 'preference'?”**
>
> This is a fair question that we will tighten our language on in the paper. We believe that preference best encapsulates our hypothesis. We’d first like to point out that the training procedures for the models samples roughly uniformly across a large protein database like UniRef50. If each sample has equivalent training emphasis, then why are some sequences presented with a lower or higher density than others after sufficient training time? Here lies the observation that the difference in observed probabilities must be a result of the data distribution, otherwise we’d expect all proteins to achieve a roughly equal likelihood.
>
> The notion of “preference” in our work is that we examine how likelihoods are affected by data composition, and data composition originates from our human preferences when we decide what to sample and include (see Ding et al., 2024 [1]). Our work demonstrates that data has a big impact on pLM likelihoods; therefore, its inability to faithfully recapitulate the fitness landscape is implicitly due to humans’ preferences in aggregating the database. This is a helpful piece of feedback in improving the clarity, and we’ve updated the writing to reflect this.
>
> **Single Inference Pseudo Likelihood**
>
> We appreciate the feedback on language within the proof and will be updating our wording with the advice as follows:
> “Under a mask-consistent masked language model that sets masked tokens [...]” will become “Under a mask-consistent masked language model **with a training scheme that** sets **training** tokens [...]”. Our language comes from the original BERT paper from Devlin et al. 2018, in which they describe the algorithm as “If the i-th token is chosen, we replace the i-th token with (1) the [MASK] token 80% of the time (2) a random token 10% of the time (3) the unchanged i-th token 10% of the time.” In addition, we will introduce our notion of “mask-consistency” earlier.
>
> **“However, what exactly is [Pseudo Likelihood …]?”**
>
> We describe the exact algorithm in Section B.1 Algorithm 1. It is consistent with the proposed calculation in the comment.
>
> **“Do the new and old approaches provide the exact same output for all possible models [...]?”**
>
> Yes (under the typical assumption in ML that models have correctly converged during training). The output is not affected by the distribution that is examined, but on how tokens were perturbed during MLM training. This is why we needed the $\alpha$ and $\beta$ terms in our algorithm.
>
> **“Given that the actual inputs to the model are different, which means that the activations are different, it appears to me that the approaches would provide numerically different outputs [...]?"**
>
> We agree with the argument presented, and would like to highlight how our proof side steps this issue. Though the model would in fact see different activations, our proof relies on the downstream probabilities instead of activations. This makes it non-reliant on these activations.
>
> Taking into account the feedback, we will make sure to make sure to emphasize the need for a model to be “sufficiently trained” in order to exhibit this capability.

---

> > ### Comment · Reviewer_NtpF · 2024-11-22
> >
> > I continue to feel that the Theorem 4.1 is far too imprecise and that the statement is not nearly formal enough to be presented as an actual Theorem (it could have been some informal remark or an informal justification for a particular approximation algorithm). I asked:
> >
> > “Do the new and old approaches provide the exact same output for all possible models [...]?”
> >
> > You responded:
> >
> > "Yes (under the typical assumption in ML that models have correctly converged during training)"
> >
> > Isn't Figure 2.1 (which is presented immediately adjacent to Theorem 4.1) an empirical demonstration that the answer should be no? If the outputs were the same, wouldn't all data be on the y = x line?
> >
> > Also, when I said 'all models', I meant 'for any value of the parameters of the model.' I don't think your definition of 'typical' for a model is that typical. What does it mean to have 'converged' during training?  Is such a condition ever achieved in practice when training big models? What converged? The loss? The parameters? Such details would matter for a theorem statement. Also, how can a statement about the dynamics of training be used to provide a guarantee about the model's outputs on held-out data, for which your theorem seems to apply?
> >
> > Your proposed solution to explain that models need to be 'sufficiently trained' does not feel satisfactory to me. If you are presenting an actual Theorem, it needs to be precise, and 'sufficiently trained' is not precise. And, most importantly, how is it sensible to present a Theorem alongside a Figure that seems to provide a counterexample to the Figure?

---

> > > ### Author Response · Authors · 2024-11-23
> > > **Response to Official Comment by Reviewer NtpF**
> > >
> > > I believe we misunderstood your initial question to be only with respect to data distribution and training dynamics, “do the new and old approaches provide the exact same output for all possible models, or only when models have maximized the likelihood of training data from a particular distribution?” Then answering above with more informal notions of loss convergence.
> > >
> > > To improve the precision of our language, we can rephrase our statements surrounding the new algorithm. Would you find it more appropriate if we reframe this as “Remark 4.1" describing the approximate relationship, with Figure 2 serving to visualize the correspondence and deviation?

---

> > > > ### Comment · Reviewer_NtpF · 2024-11-25
> > > >
> > > > I'm glad we have converged to a consensus that this observation about the model's behavior should be presented as an informal remark that motivates an approximation algorithm instead of a formal statement, as this would depend on a significant number of assumptions (training distribution, test distribution, train loss, training dynamics, etc) that seems to be violated in Figure 2.
> > > >
> > > > IIUC, you are allowed to post additional comments between now and Nov 27th. Can you please please write up the new version of the 'Remark' in its full text and include it in the response?

---

> > > > > ### Author Response · Authors · 2024-11-26
> > > > > **Response to Official Comment by Reviewer NtpF**
> > > > >
> > > > > In our revision, we will update Sections 4.1 and 4.2 with the following language. It now incorporates the feedback above surrounding describing the masking, defining mask-consistent earlier, and rephrases our algorithm plus result as an approximation rather than a full theorem (as stated above most assumptions would be violated in practice).
> > > > >
> > > > > # Section 4.1
> > > > >
> > > > > [...]
> > > > >
> > > > > We argue that PLL can be approximated for any masked language model in a single forward pass using Algorithm 2 as a result of Remark 4.1. To achieve this, we define the term “mask-consistent”, where a token's masked probability is equal to its probability under the implicit assumption that the token might have been scrambled by the training procedure written $P(y_i = x_i | x, \phi_i=1) = P(y_i = x_i | x_{\setminus i})$. Using mask consistency, we produce a way to approximate the probability of a masked token when the token is not masked during inference in Remark 4.1, then justifying it in Section 4.2. As a result, we can approximate PLL for any model without requiring bespoke finetuning or exhaustive resources.
> > > > >
> > > > > ### Remark 4.1
> > > > > Under a mask-consistent masked language model with a training scheme that sets training tokens to a random token with probability $\alpha$ and keeps them unchanged with probability $\beta$: $P(y_i = x_i |x_{\setminus i}, \theta) = \frac{\alpha + \beta}{\alpha} P(y_i = x_i |x, \theta) − \frac{\beta}{\alpha}$

---

> ### Author Response · Authors · 2024-11-21
> **Response to Reviewer NtpF Part 2**
>
> **“Also, why is the single-pass approach actually necessary in your experiments?”**
>
> PLL has been an important metric within the pLM community with its extensive use within ESM-2 [4] and as a validation metric in protein sequence generation works. This mirrors the commonly used perplexity metric in NLP to assess characteristics of language models; since the ESM series of models are trained with the masked-language modeling objective, existing literature uses this O(L) variant. Please see Supplementary Materials, Appendix 2.2 in Lin et al., 2024 [5]. With an average length of 300 residues in these databases, our algorithm provides a 300x speedup in wall clock time democratizing the use of this computation.
>
> To put this speedup into perspective, let's compare calculating PLL for a protein database to the costs of training a pLM from scratch. ESM-2 650M was trained for roughly 25 epochs (2M tokens per batch / 300 residues per protein * 270K steps / 65M sequences in UniRef50 = 25 epochs) [5]. Calculating PLL on all of these sequences at 300 residues per protein would be 300 inference passes over the entire dataset, using the estimation from Kaplan et al. [6] that a full training step is computationally equivalent to 3 inference passes, means that this would be the same as training for 100 epochs. In sum, if any researcher wants to calculate this statistic for a large database they need 4 times the resources required during the training of a modern foundation model.
>
> **“The paper focuses on DMS datasets that measure different attributes of a protein [...] yet it uses assay-independent models.”**
>
> We agree that this does upper bound performance, but this is the norm for studying zero-shot fitness prediction like in Meier et al. [3] and ProteinGym [4].
>
> **“What if, for example, the correlations you are seeing are due to the fact that low-likelihood proteins are from the sort of species where the DMS datasets are based on high-level observations about organismal survival, instead of low-level molecular biology measurements like binding?”**
>
> This motivates a good follow up experiment to disentangle this effect. We’ll recreate the panels in Figure 2, but split by selection criteria, then recalculate Spearman correlations to ensure that this trend holds over different families.
>
> **“I think influence functions are cool and I enjoyed seeing that you used them to study protein LMs, but I don't think the inclusion of the influence function results drives the paper's story enough to be included.”**
>
> Although our results reinforce a previous method, we found it important to increase the thoroughness of the paper and used the results to motivate our intervention. Influence functions provide a specific mechanism to study how training data impacts our feature of interest: likelihood.
>
> **“Can you please explain the connection to Qin and Colwell in more detail?”**
>
> Qin and Colwell observe a power law tail over the eigenvalues of the covariance matrix and we find one over the distribution of influence values. Moreover, Qin and Colwell found that the power tail signal harmed performance in parallel to how our high likelihood sequences perform poorly.  This provides evidence towards the results of Weinstein et al. 2022 [7] where phylogenetic signals corrupts fitness signals. We will update the paper to better convey this.
>
> **“Can you elaborate on the result of sec 5.2? How is this observation actionable? How does it inform the other parts of the paper?”**
>
> Because sequence overlap improves likelihood most, we take the lowest E-value sequences as they would have the most overlap with the wild type during evo-tuning.

---

> ### Author Response · Authors · 2024-11-21
> **Response to Reviewer NtpF Part 3**
>
> **“I didn't understand figure 1B. [...] Why would the overall likelihood of the wildtype be correlated with the size of this set?”**
>
> The fluctuation in the size of plausible mutations varies with likelihood as the likelihood of wild type gives information about how much uncertainty the model has about residues. In particular, as the model gets more confident (higher likelihood) the number of mutations that are likely to be sampled goes down. When the log likelihood is zero, the model expects there to be no variation in the protein sequence at all. This figure extends Figure 1 of Hie et al. 2023 [8] adding more depth to the proposed hypothesis of efficient evolution.
>
> [1] Ding et al. 2024, “Protein language models are biased by unequal sequence sampling across the tree of life”
> [2] Devlin et al. 2018, “BERT: Pre-training of Deep Bidirectional Transformers for
> Language Understanding”
> [3] Meier et al. 2021, “Language models enable zero-shot prediction of the effects of mutations on protein function”
> [4] Notion et al. 2023, “ProteinGym: Large-Scale Benchmarks for Protein Fitness Prediction and Design”
> [5] Lin et al. 2023, “Evolutionary-scale prediction of atomic-level protein structure with a language model”
> [6] Kaplan et al. 2020, “Scaling Laws for Neural Language Models”
> [7] Weinstein et al. 2022, “Non-identifiability and the blessings of misspecification in models of molecular fitness.”
> [8] Hie et al. 2023, “Efficient evolution of human antibodies from general protein language models”

---

> ### Author Response · Authors · 2024-12-02
> **Deadline Reminder**
>
> We thank the reviewer for their constructive feedback and engagement throughout the discussion period. Given that the comment period closes tonight, we would appreciate the reviewer’s thoughts on whether our responses and clarifications have addressed their concerns regarding the initial score. If there are any remaining points we can clarify to help with this assessment, we would be happy to address them promptly.

---

### Official Review · Reviewer_fqUe · 2024-11-04

**Soundness:** 3
**Presentation:** 3
**Contribution:** 4
**Rating:** 8
**Confidence:** 4

**Summary:**

This paper investigates the effectiveness of Protein Language Models (pLMs) in predicting protein fitness in a zero-shot setting. In essence, it explores whether pLMs can capture treu biological patterns or artifacts of the training data. It analyzes hundreds of deep mutational scans through the likelihood-based preference and finds that both overly high and overly low sequence likelihoods decrease prediction accuracy. It further leverages influence functions to trace the origins of these likelihoods and proposes "evo-tuning" as a remedy to improve pLM performance on low-likelihood proteins.

**Strengths:**

This paper introduces a novel perspective by framing the predictive capacity of pLMs as a "preference" rooted in sequence likelihood. This framing is innovative, though this kind of framing is already common in RLHF for LLM.

It is among the first to rigorously employ influence functions to trace the impact of training data on pLM predictions, effectively combining causal inference with protein sequence modeling.

There are too many methods and tricks for improving performance on different kinds of datasets which simply apply most state-of-the-art language model methods and tricks into protein but lack more careful inspection into the problem. This paper thinks more deeply into the question and carefully examine the pLMs through implicit preference.

**Weaknesses:**

The experiment would be better if it includes more datasets to fully support some claims, for example, evo-tuning pLMs could improve fitness prediction.
Also, I am wondering if Bert-like esm-3 or other smaller Bert-like pLMs may be examined.

**Questions:**

Please refer to the weaknesses.

---

> ### Author Response · Authors · 2024-11-21
> **Response to Reviewer fqUe**
>
> We’re grateful for the positive reception towards our problem framing and simplicity. As noted in the review, there are many complicated methods that lead to marginal performance gains, so it was our aim to provide a simple well-motivated intervention.
>
> **“The experiment would be better if it includes more datasets to fully support some claims [...]”**
>
> While we agree that more experiments would add more confidence, we ran 217 unique different evo-tuning experiments to draw our conclusions and reinforce our findings.
>
> **“Also, I am wondering if Bert-like esm-3 or other smaller Bert-like pLMs may be examined.”**
>
> This is a good point, we’ll aim to include more results for ESM-3 [1]  and ProtBert [2] in the camera-ready version. One minor technical note is that the substitution parameters for ESM-3 aren’t explicitly given in the work, so estimating PLL in a single pass will require us reaching out to the authors.
>
> [1] Hayes et al. 2024, “Simulating 500 million years of evolution with a language model”
> [2] Elnaggar et al. 2021, “ProtTrans: Towards Cracking the Language of Life’s Code Through Self-Supervised Learning”

---

> > ### Comment · Reviewer_fqUe · 2024-11-27
> >
> > Thank you for the reply. I will keep my score as it is already a very positive score. I am looking forward to see more pLMs examined in the future version.

---

### Author Response · Authors · 2024-11-21
**Response to all Reviewers**

We’d like to thank reviewers for their time spent engaging with the work. We appreciate the positive feedback towards our work providing a new perspective on understanding fitness capabilities and its simple solution.

The in-depth comments provided by reviewers have informed improvements to the work. We respond in detail to each reviewer's questions and feedback in a respective comment.

---

### Meta-Review · Area_Chair_oKpJ · 2024-12-18

**Metareview:**

Protein language models (pLMs) trained on evolutionary data show promise in designing functional proteins but face challenges in predicting when they will succeed in zero-shot fitness estimation. This submission reveals that the likelihood of a protein sequence, influenced by pretraining preferences and sequence homology, predicts pLM performance, with over- or under-preferred sequences harming outcomes. Low-likelihood sequences can be improved through unsupervised fine-tuning, offering guidance on when to deploy pLMs in protein engineering and how to enhance their performance in challenging scenarios.

Most reviewers acknowledge the innovation and comprehensive experimental validations of the proposed methods. However, as one of the reviewers pointed out there are several issues in the paper presentation and organization. During the rebuttal, the authors have addressed most concerns. Meanwhile, we highly encourage the authors to update the draft accordingly. We recommend an acceptance.

**Additional Comments On Reviewer Discussion:**

During the rebuttal, the authors have addressed most concerns. Meanwhile, we highly encourage the authors to update the draft accordingly.

---

### Decision · Program_Chairs · 2025-01-22

Accept (Poster)